# Allylamine PECVD Modification of PDMS as Simple Method to Obtain Conductive Flexible Polypyrrole Thin Films

**DOI:** 10.3390/polym11122108

**Published:** 2019-12-15

**Authors:** Robert Texidó, Salvador Borrós

**Affiliations:** Grup d’Enginyeria de Materials (GEMAT), Institut Químic de Sarrià, Universitat Ramon Llull, Via Augusta, 390, 08017 Barcelona, Spain; roberttexidob@iqs.url.edu

**Keywords:** conducting polymers (CPs), conjugated polymers, stretchable electronics, surface modification, plasma grafting

## Abstract

In this paper, we report a one-step method to obtain conductive polypyrrole thin films on flexible substrates. To do this, substrates were modified through allylamine plasma grafting to create a high amount of reactive amine groups on PDMS surface. These groups are used during polypyrrole particle synthesis as anchoring points to immobilize the polymeric chains on the substrate during polymerization. Surface morphology of polypyrrole thin films are modified, tailoring the polyelectrolyte used in the polypyrrole synthesis obtaining different shapes of nanoparticles that conform to the film. Depending on the polyelectrolyte molecular weight, the shape of polypyrrole particles go from globular (500 nm diameter) to a more constructed and elongated shape. The films obtained with this methodology reflected great stability under simple bending as well as good conductivity values (between 2.2 ± 0.7 S/m to 5.6 ± 0.2 S/cm).

## 1. Introduction

Considering the importance of electronic devices nowadays, is undeniable that providing new properties to electronic materials would open the door to new devices with a high impact in our lives. The development of electronic materials that avoid the stiffness of silicon and aim to enhance flexibility and stretchability of electronic devices has experienced an increasing interest in recent years, expanding the range of applications [1,2,3]. An example of this can be found in smart textiles [4,5,6], electronic skin [7,8,9], wearable devices for medical diagnosis [10,11], bioelectronic implants [12,13] or soft robotics [14]—among others. Conducting polymers (CPs) play an unique role in the development of these new electronic devices due to their extraordinary advantages over traditional materials: tunable conductivity, low density, high stability, low cost, shape and morphology control, corrosion resistance, and chemical diversity, when compared with their inorganic counterparts [15,16,17]. At present, CPs, including polythiophene (PTh), polypyrrole (PPy), and polyaniline (PANI), are widely studied to be implemented in flexible electronic devices, simultaneously trying to solve their main drawback: processability. 

Extended π electron delocalization present in CPs backbone allows charge transportation but provides rigidity to polymeric chains. This phenomenon causes CPs to be insoluble in most organic solvents and to have low adaptability to substrates, decreasing mechanical deformability and robustness [18,19]. All of the above dramatically limits the implementation of conducting polymers, especially in flexible applications. Although considerable efforts have been made, current strategies to provide flexibility to CPs do not fulfill the requirements of the expected applications, especially in the biomedical field, or are not suitable for industrial scale up. 

Considering relevant techniques to process CPs (electropolymerization, vapor phase polymerization or electrospinning, among others) only few are compatible with flexible substrates. Electropolymerization allows for a high degree of control on electrochemical properties of films [20,21] but polymer deposition on flexible substrates requires a flexible working electrode, which is an unsettled issue [22]. Electrospinning CPs allows for a certain degree of flexibility, producing CPs fibers whose structure can be easily adapted to substrate morphology [23,24,25]. However, fabrication of small components with controlled geometries is complex when using electrospinning techniques, and makes scalability very challenging. Other strategies to provide flexibility to CPs consist in the modification of monomers, polyelectrolyte or other elements of the polymerization media to enhance the adaptability with the substrate. For example, some studies show that dopamine based polyelectrolytes allow to improve adhesion and conductivity of polypyrrole films [26,27]. In a similar way, it acts as part of the utilization of plasticizers such as Zonyl to form films with higher crack-onset strains on flexible substrates [28,29]. Even though some strategies may represent a good solution to solve the scale up challenge of CPs, further studies must be performed to validate that these modifications on polymer structure do not affect other properties such as stability or long-term conductivity. 

In this context, vapor phase polymerization, chemical vapor deposition (CVD), and oxidative chemical vapor deposition (oCVD) processes mimic the step growth polymerization. However, in the vapor phase, eliminating problems in CPs processing such as monomer solubility or substrate wettability allowing us to obtain thin films of CPs in a wide range of flexible substrates [30,31,32,33]. Despite all the advantages that CVDs techniques provide, they also present several drawbacks. Elements such as vacuum chambers or systems to vaporize and introduce reagents in vapor phase require a complex set up that reduces scalability and does not allow for the manufacturing of multilayer electronic devices composed of more than one material [34,35,36]. Moreover, intrinsic to CVD technologies, monomers and reagents used to obtain CPs films are limited by the technique requirements, such as the ability of the monomers to be evaporated. These limitations do not only reduce the applicability range but also prevent the use of liquid phase polymerization strategies to tailor some film parameters such as morphology, which plays a critical role in the electric properties [37]. 

Thus, it seems interesting to explore mixed strategies that combine the advantages of CVD techniques with the versatility of liquid phase polymerization to obtain films of flexible conductive polymers. In this sense, modification and functionalization of substrate can enhance the interactions between monomer and substrate assisting the deposition of the film. This paper presents how performing a simple surface modification on a flexible substrate through plasma enhanced vapor deposition (PECVD) treatment a CP thin film can be immobilized. Thereby, this technique avoids the complexity of CVD techniques maintaining its applicability on large substrates and allowing liquid phase polymerization strategies. Specifically, in this work it is demonstrated that the surface of polymethyl siloxane (PDMS), known as the gold standard of stretchable materials, can be modified with amine groups through PECVD. Exposure of PDMS to argon oxygen plasma allows for the formation of free radicals on the surface. These highly reactive sites allow vaporized molecules to be grafted, as in the case of allylamine, for the functionalization of PDMS surface with amine groups. 

After functionalization, modified PDMS allows for the immobilization of polypyrrole (PPy) nanoparticles during its synthesis, obtaining a polypyrrole (PPy) thin film. The utilization of a polyelectrolyte during PPy particle formation played a critical role enhancing the interactions between monomer and modified surface, obtaining thin flexible conductive PPy films. Film conductivity and morphology of the film has been adjusted modifying the polyelectrolyte during PPy particle formation and will be assessed in the present work. 

## 2. Materials and Methods 

### 2.1. Modified PDMS—Allylamine Grafting through PECVD and Plasma Reactor

Allylamine grafting was carried out using a stainless-steel vertical plasma reactor manufactured by the GEMAT group (Barcelona, Spain) and previously described (Appendix A) [38]. To perform allylamine grafting modification, the first step consists in plasma activation, where a mixture of oxygen and argon (80:20 v/v) was fed until a working pressure of 0.14 mbar was achieved for the desired experimental time (5 min). After that, gas mixture was cut off and power generator was turned off to stop the formation of radicals on the substrate. Then, the monomer (allylamine purchased from Sigma Aldrich, San Louis, MO, US) was regulated through the needle valve and introduced into the reactor chamber in vapor phase (heated at 40 °C) until 0.15 mbar for the desired experimental time (15 min). Optimization of allylamine grafting conditions are presented in Table 1. 

Unless otherwise stated, all experiments in the plasma treatment section were done by triplicate. Average mean and standard deviation of triplicates were represented after the treatment, when the monomer or gases valves were closed, vacuum was cut off, and samples were removed from the reactor and stored with argon atmosphere until further use.

### 2.2. Polypyrrole Nanoparticle Synthesis

Reagents used in the polypyrrole nanosuspension synthesis consisted of pyrrole (≥99%, Aldrich, San Louis, MO, US), hydrochloric acid (37% purchased from Sigma Aldrich, San Louis, MO, US), ammonium persulfate (≥98%, Fluka, Bucharest, Romania), poly (sodium 4-styrenesulfonate) solution (Mw 70,000, 30% wt) and (Mw 200,000, 30% wt) purchased from Sigma Aldrich (San Louis, MO, US).

For the synthesis of PPy:PSS, pyrrole monomer (0.053 mmol) was introduced dropwise in 40 mL of HCl 0.03 M. The solution was stirred vigorously for 1 h and then concentration 3 × 10^−4^ mMol of PSS solution was added dropwise. After 1 h, 5 mL of a 6 × 10^−3^ M ammonium persulfate solution was added to the mixture as oxidizing agent to start pyrrole polymerization for 12 h at room temperature. After this time, a characteristic black coloured suspension was obtained.

### 2.3. Nanosuspension Characterization

The particle size of the nanosuspensions was determined by dynamic light scattering in diluted samples (1:10) using a Zetasizer ZEN3600 (Malvern Panalytical, Malvern, United Kingdom).

### 2.4. Microscopy Images

Scanning Electronic Microscopy (SEM) images were taken to observe the nanostructure with a field-emission Zeiss Merlin FE SEM (Oberkochen, Germany). The observed samples were drop casted (200 µL) on a silicon wafer dried at room temperature. PDMS modified samples with PPy thin film were observed using the same microscopy directly.

### 2.5. Conductivity Characterization

Electrical conductivity measurements were performed using four-point probe set up with Keithley Sourcemeter 2600 (Keithley, Beaverton, Oregon, US) with a spatial positioner to fix the electrode to the sample. To calculate the conductivity, the thickness of the films was measured with a confocal microscope Leica DCM 3D with 50x objective (NA 0.90) (Leica, Wetzlar, Germany).

### 2.6. Substrate Preparation

PDMS substrate were prepared using Sylkgard 184 silicone kit (Dow Corning, Midland, Michigan, USA). The two components were mixed in a silicone/initiator ratio of 10:1 (v/v). Then, the mixture was uniformly deposited on a glass plaque with a manual film applicator (500 µm thickness) and then cured at 150 °C for 6 h.

## 3. Results

### 3.1. Allylamine Grafting through PECVD

First, PDMS substrates were functionalized with amine groups through PECVD allylamine grafting. The main strategy consists in promoting the interaction between amino groups on the surface and pyrrole during liquid phase polymerization. Therefore, modified PDMS has been evaluated as a platform where a PPy thin film can be immobilized by placing the substrate into polymerization media. Schematics of surface modification of PDMS are shown in Figure 1. Allylamine plasma grafting was performed in two steps. First, PDMS substrates were exposed to plasma of a mixture of oxygen and argon forming unstable groups and free radicals on the surface of the substrate in contact. In the second step, allylamine was vaporized into the reactor chamber to get in contact with the free radical sites. These free radicals are able to react with vinyl based monomers, in this case vaporized allylamine monomer, to create a “brush-like” structure with oriented amine groups [39].

Allylamine grafting was performed on PDMS substrate using the conditions and procedures described in materials and methods section. To validate that PDMS was correctly modified obtaining a surface “rich” in amine groups after the PECVD, surface characterization through IR spectra and water contact angle (WCA) was performed. IR-ATR confirmed the existence of amine groups on PDMS surface after the allylamine grafting, as shown in Figure 2A. After the allylamine grafting, new small absorption bands appeared on the IR-ATR spectra of the modified PDMS. The wide absorption band at around 3100 to 3500 cm^−1^ (green zone) may be attributed to stretching vibrations of N-H bonds in amine [40]. Small shoulders bands at 1633 and 1564 cm^−1^ (blue zone) may be associated with the deformation bending of the -NH_2_ group [41]. In order to complement IR-ATR spectra to verify that PDMS surface has been functionalized with amine groups, WCA measurements were performed.

WCA is a technique that allows to measure the angle formed by a drop of water on the surface of the substrate revealing the wettability of the solid surface. We used WCA measurements of allylamine modified PDMS before and after the modification (Figure 2B,C) to evaluate changes in surface wettability after the allylamine grafting. WCA varies from 90.2 ± 0,1° for non-modified PDMS surface to 64.6 ± 0,2° for modified PDMS. This change in WCA matches with values reported in bibliography for NH_2_ functionalized surfaces [42]. These values combined with the new bands observed in IR-ATR spectra indicate that PDMS has been successfully functionalized. 

### 3.2. Polypyrrole Nanosuspension Prepared through Electrostatic Interaction Synthesis 

The immobilization of PPy/PPy:PSS on modified PDMS was studied by liquid phase polymerization. For this purpose, substrates were placed inside the polymerization medium of PPy/PPy:PSS particles. With this in mind, we studied the synthesis of PPy nanosuspension through electrostatic interaction synthesis. This methodology consists of the use of electrostatic forces between monomer and a polyelectrolyte during oxidative polymerization to control particle size. The polyelectrolyte has a similar effect to that of the micelle structures, isolating the monomer during radical polymerization and controlling the chain growth, obtaining narrower particle size distribution. Poly(sodium 4-styrenesulfonate) (PSS) polyelectrolyte of two different molecular weights (Mw 70,000 PSS 1 and Mw 200,000 PSS 2) was used in this study to obtain PPy:PSS stable nanosuspensions. PSS plays two interesting roles: first, it acts as a counter anion for PPy conductivity. Secondly, it increases the interaction with the amine groups on PDMS surface. 

PSS is widely used in the synthesis of polyaniline:PSS nanosuspensions through electrostatic interaction synthesis [43,44,45] but there is not much bibliography describing the same procedure for pyrrole. Thus, the obtained PPy:PSS nanosuspension morphology, size distribution and film conductivity were characterized. 

PPy:PSS nanosuspensions were synthetized as described in the materials and methods section. After polymerization, a black coloured solution characteristic of polypyrrole stable suspensions was obtained (Figure 3A). FE-SEM images of PPy:PSS particles (Figure 3B,C) show a globular morphology in the deposited particles in the nano-range, being similar in size to other PPy particles [46]. Figure 3D reveals that when PSS of higher molecular weight is used, the nanoparticle obtained presents a more elongated shape. This effect may be attributed to the conformation in which PSS surrounds the pyrrole monomer during polymerization. Smaller polyelectrolyte molecular weights would limit the pyrrole chain growth, resulting in small globular particles while high molecular weight would present more space for pyrrole growth, driving to elongated particle form. These changes in PPy:PSS nanoparticles reveal the key role that polyelectrolyte plays during polymerization as well as an interesting way to control the morphology. DLS measurements confirmed that obtained PPy:PSS presents an homogeneous size distribution under 300 nm for both molecular weights of PSS with a mean value of 297 nm for PSS Mw 70,000 and 195 nm for PSS Mw 200,000 (Figure 3D) with a PDI near 0.2 in both cases. The electrical conductivity of the PPy:PSS films on a silicon wafer (Figure 3E) for the two synthetized nanosuspensions presented values in the range of polypyrrole conductivity films [47,48]. As expected, when a PSS of higher molecular weight is used during PPy:PSS synthesis, the conductivity value increases, most likely due to the doping effect of the extra sulfonate groups [49]. The characterization of PP nanosuspensions reflects the potential of nanosuspensions as a method to obtain conductive films with easily modifiable properties.

### 3.3. One-step PPy Nanoparticle Inmmobilization 

In this section PDMS modified substrates have been evaluated as a platform to immobilize PPy/PPy:PSS nanoparticles. To do this, substrates were placed in the polymerization media during the electrostatic interaction synthesis of PPy/PPy:PSS.

As expected, attempts to deposit PPy:PSS nanosuspension directly on unmodified PDMS by drop casting resulted in very little material being maintained in the substrate. Moreover, the few PPy:PSS nanoparticles adhered to the substrate form a very cracked film. This effect is due to the low wettability of PDMS combined with the few interactions of PPy chains with the substrate. Simple plasma treatments on PDMS surfaces, such as activation, improve substrate wettability but the lack of surface interactions between surface and film still produces intense cracks on film (Appendix A). To avoid this behaviour, the enhancement of interactions between pyrrole, PSS and amine groups to immobilize polypyrrole thin films has been studied (Figure 4). Deposition of PPy thin films on the modified PDMS surface was performed by placing the substrates inside a petri dish before pyrrole polymerization. Immediately, an oxidizing agent was added to the media to start the polymerization. Our assumption is that when liquid phase polymerization occurs, the monomer, polyelectrolyte and surface amino groups interact with each other, promoting interactions between polymeric chains and surface. To do so, amine groups present in modified PDMS play different roles to allow this assisted immobilization. First, amine groups form electrostatic interactions with pyrrole approaching monomers to the surface during the growth step process. If electrostatic interactions are strong enough, pyrrole growing chains can be also immobilized by physical interactions inside PDMS surface moieties. At last, amine groups will also create electrostatic interactions with polyelectrolytes present in the polymerization media, allowing the immobilization of entire polypyrrole nanoparticles during their formation. 

Different nanosuspension media were evaluated to immobilize PPy:PSS nanoparticles in the PDMS surface by the interaction of the components formed during the synthesis. The difference between media is the employed polyelectrolyte (Table 2). Evaluated nanosuspensions were synthetized using the conditions described in the materials and methods section, incubated in contact with PDMS surface for 12 h and vigorously cleaned with water. Submersion of unmodified PDMS or plasma treated PDMS on PPy:PSS nanosuspension media during nanoparticle polymerization did not produce film formation on the surface of PDMS. Optical microscopy images of PDMS after incubation on the nanosuspension media revealed no new visible structures that may be attributed to polypyrrole nanoparticle cluster or polymer deposition phenomenon. IR spectra of PDMS after incubation do not show any differences with raw PDMS IR spectra, confirming that no nanoparticle formed from the nanosuspension synthesis has been immobilized.

However, when allylamine modified PDMS was examined after being incubated on PPy:PSS nanosuspensions, it presented significant differences when compared to the unmodified PDMS. Visual inspection and optical microscopy images of the samples after incubation showed that a continuous polymeric film had been formed on the substrate. 

The comparation of IR spectra between Allylamine modified PDMS and unmodified PDMS samples after the incubation revealed the apparition of new bands, which can be attributed to PPy:PSS immobilization. The band at 3440 cm^−1^ corresponding with N-H stretch [50] (Figure 5A), the band at 1550 cm^−1^ corresponding with a typical pyrrole ring vibration and the band at 1280 cm^−1^ corresponding with the C-N stretch band of secondary amine, (Figure 5B) indicate the presence of PPy in the surface [51,52]. All band intensities observed in the IR spectra are higher than those of unmodified PDMS or allylamine modified PDMS controls, remarking that IR bands correspond to new bonded molecules on the surface of the films and confirming that allylamine modified PDMS surfaces work as a platform where molecules can be immobilized.

To study the morphology of the PPy/PPy:PSS structures deposited on allylamine modified PDMS, FE-SEM images of the films were taken for each formulation. Figure 6A reveals the morphology of samples incubated during the polymerization of pyrrole without PSS. Extended images show that the film was composed by PPy globular nanoparticles (Figure 6B). It is interesting to highlight that direct oxidative polymerization of pyrrole in acid media without PSS produces a black dark precipitate. Instead of that, PPy nanoparticles of near 100 nm seem to be adhered to the allylamine modified PDMS. These nanoparticles present a highly compact morphology, which seems to be the result of a coalescence process that merged the particles forming a continuous film (Figure 6B). 

By contrast, samples where modified PDMS was incubated with PSS during pyrrole polymerization revealed a different morphology, as it is the case of Al–PPy:PSS 1 (PSS 70,000) and Al–PPy:PSS 2 (PSS 200,000) (Figure 6C,E respectively). In both cases, a continuous multilayer film formed by nanoparticles can be observed. Figure 6D, where a PSS of lower molecular weight was used, show a homogeneous distribution of the PPy:PSS nanoparticles of nearly 200 nm of diameter. When a higher PSS molecular weight was used (Figure 6F), PPy:PSS nanoparticles presented an elongated tubular shape almost 500 nm of length and 200 nm of diameter. It can be concluded that by using polyelectrolytes of different Mw, it is possible to control particle size and morphology. This fact could be of high interest in applications where CPs surface must present a specific interphase. Finally, we would like to highlight that the morphology study carried out not only confirms that pyrrole has been immobilized on PDMS surface, but also that it forms a continuous film in each studied sample.

The developed films flexibility can be seen in Figure 7A–C. In this sequence of images, it is shown that the thin film is adapted at bending without observing film delamination. This behavior can be explained through the morphology images, where it has been observed how the PPy and PPy:PSS nanoparticles have been adapted very well onto the surface of allylamine modified PDMS. Interactions created with allylamine grafted molecules in a brush-like structure allow the degree of adaptability of PPy and PPy:PSS nanoparticles. The film conductivity—perhaps the most important property of CPs thin films—was evaluated after the manual stretching (Figure 7D). Conductivities obtained for the studied samples were 2.2 ± 0.7 S/m for Al–PPy, 0.056 ± 0.02 S/m for Al–PPy:PSS 1 and 1.12 ± 0.6 for Al–PPy:PSS 2, as it is shown in Figure 7D. These conductivities differ dramatically from those of the raw PDMS material or allylamine modified PDMS, for which conductivity values are in the order of 10^−9^ S/m, indicating that practically no electronic conductivity exists. Instead of this, PPy:PSS thin films conductivity values are in the range of PPy CPs [46,53], which are very similar to the conductivity values of PPy:PSS nanosuspensions deposited by drop casting presented in Section 3.2. Thus, the conductivity values obtained allow PPy:PSS thin films deposited on PDMS to be used in a wide range of applications where flexible conductive films would be needed.

## 4. Conclusions

In this study, an easy one-step method to deposit PPy/PPy:PSS conductive thin films on stretchable PDMS substrates has been presented. To do so, modified PDMS with allylamine PECVD was incubated inside pyrrole polymerization. The interactions between amine groups present in the surface and monomer/polyelectrolyte allow the assisted immobilization of PPy nanoparticles on the PDMS surface to form a continuous thin film. The morphology of these films can be controlled through the synthesis media of the pyrrole nanosuspensions. The ability to modify the size and the uniformity of the deposited nanoparticles is a powerful tool for applications were surface interaction with liquids plays a critical role, such as in biosensors field. Furthermore, conductivity values of PPy:PSS thin films are in the range of CPs—even after bending. Future work will be devoted to the study of how mechanical deformation impacts on electrical conductivity performance and how these films could be implemented in more complex electronic devices.

## Figures and Tables

**Figure 1 polymers-11-02108-f001:**
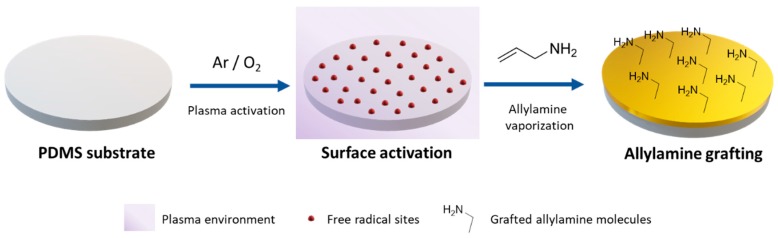
Schematics of allylamine grafting modification of PDMS substrates through PECVD. First, an active carrier gas in contact with the substrate forms radical groups in the surface of PDMS. Afterwards, vaporized allylamine reacts with radical groups, forming an amine modified surface.

**Figure 2 polymers-11-02108-f002:**
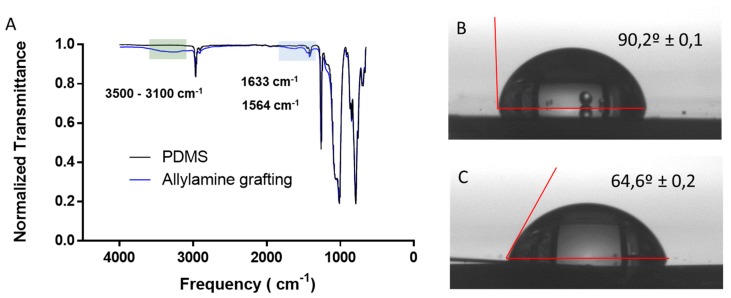
PDMS-Allylamine grafting modified characterization. (**A**) IR-ATR spectrum of PDMS and Allylamine grafting PDMS modified; (**B**,**C**) water contact angle of a water drop on PDMS before and after the allylamine plasma grafting.

**Figure 3 polymers-11-02108-f003:**
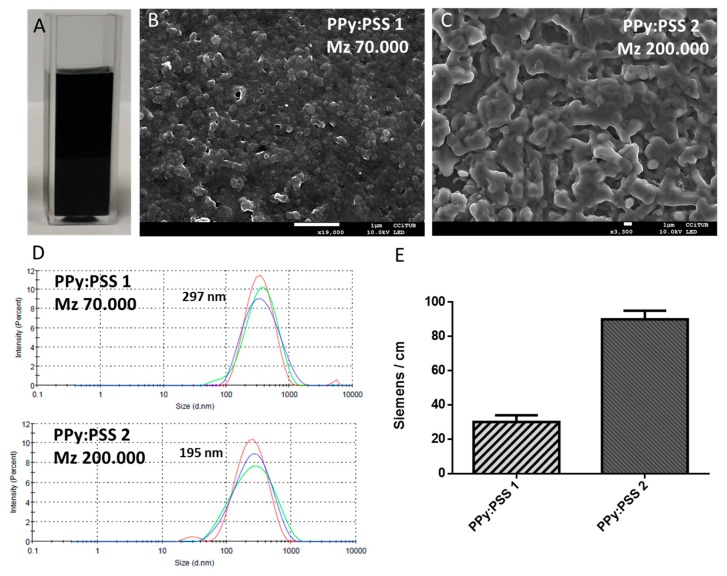
PPy nanosuspension characterization. (**A**) PPy nanosuspension. FE-SEM images of PPy:PSS nanosuspension prepared with PSS of two molecular weight; (**B**) Mw 70,000 and (**C**) Mw 200,000. (**D**) DLS size distribution of PPy:PSS nanosuspensions. (**E**) Conductivity values of PPy:PSS nanosuspension deposited by drop casting on silicon wafer.

**Figure 4 polymers-11-02108-f004:**
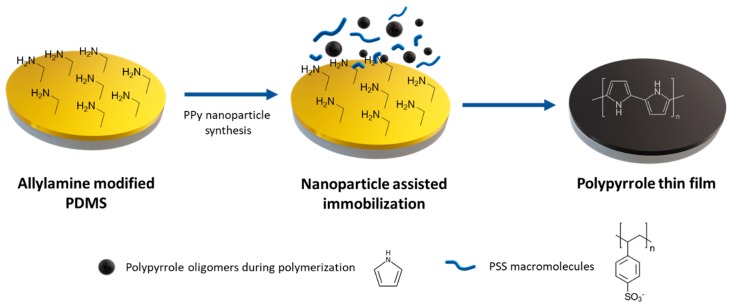
PPy:PSS nanoparticles immobilization on allylamine modified PDMS scheme.

**Figure 5 polymers-11-02108-f005:**
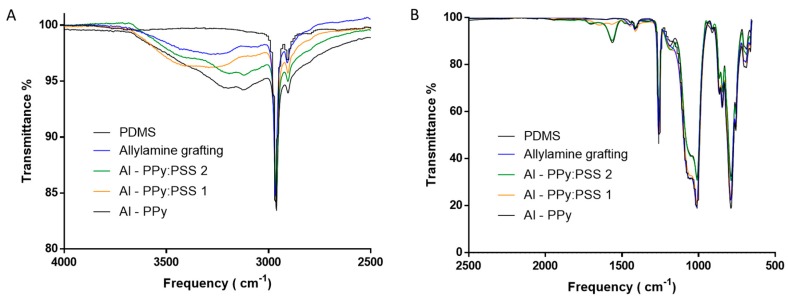
Allylamine modified PDMS films after incubation. Evaluation of nanosuspension adhesion: IR spectra of Allylamine modified PDMS after incubation (**A**,**B**).

**Figure 6 polymers-11-02108-f006:**
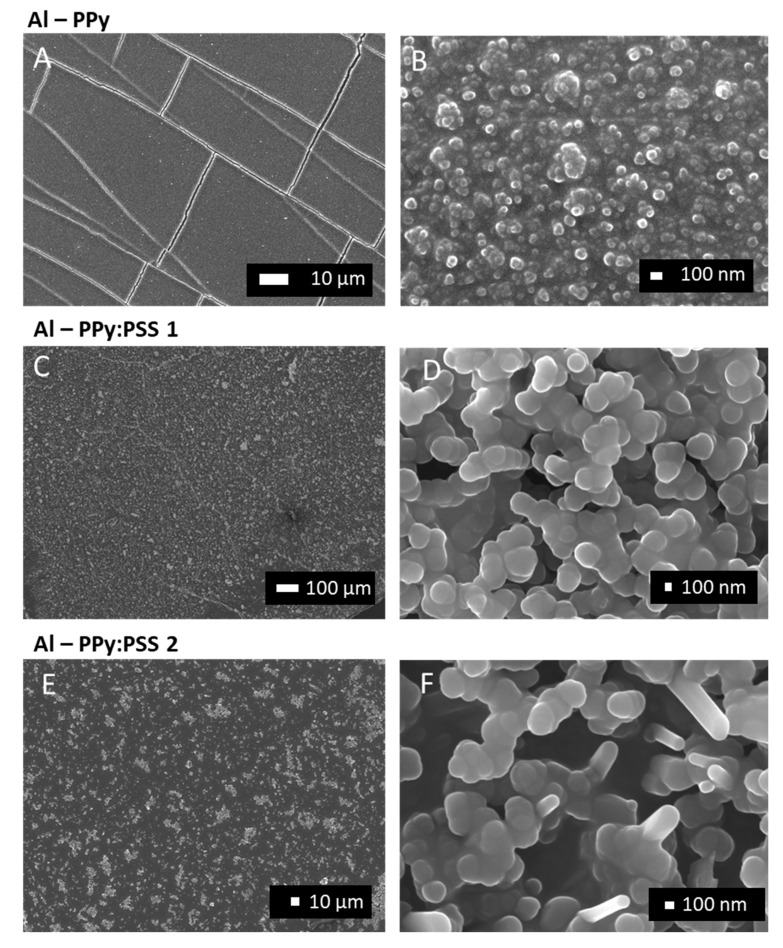
FE-SEM images of allylamine modified PDMS after nanosuspension immobilization at different magnifications: Incubated only with pyrrole (Al–PPy) (**A**,**B**); incubated with pyrrole and PSS Mw 70,000 (Al–PPy:PSS 1) (**C**,**D**); incubated with pyrrole and PSS Mw 200,000 (Al–PPy:PSS 2) (**E**,**F**).

**Figure 7 polymers-11-02108-f007:**
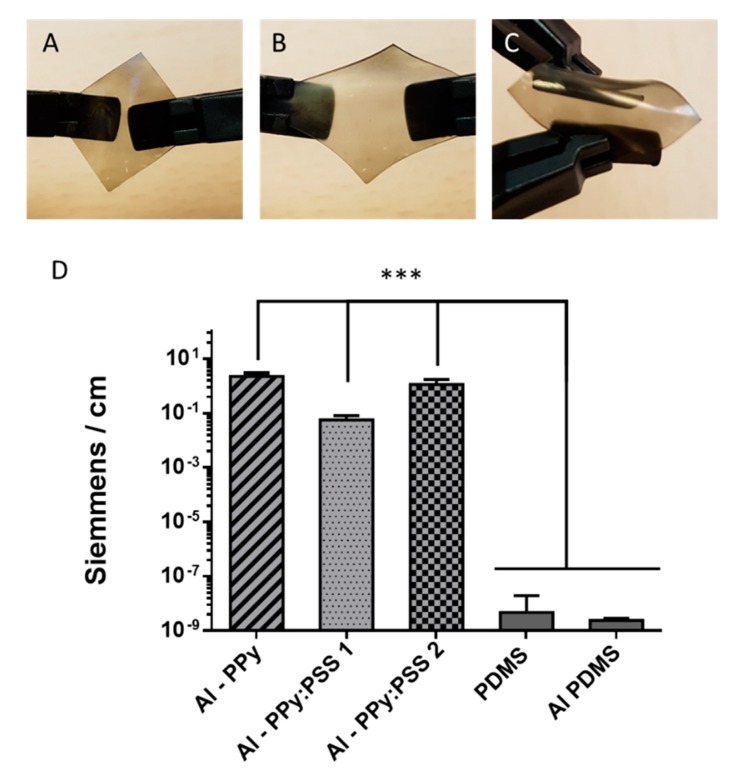
Images of flexibility behaviour of PPy:PSS films formed on Alylamine modified PDMS (**A**–**C**) and conductivity behavior of the films after stretch (**D**).

**Table 1 polymers-11-02108-t001:** Conditions of allylamine grafting through PECVD for PDMS substrate modification.

Plasma Activation		Allylamine Grafting	
**Used Gas**	**O_2_/Argon**	Vaporization Temperature	40 °C
Proportion	80:20	Time	15 min
Gas pressure	0.14 mbar		
Power	25 W		
Activation time	5 min		

**Table 2 polymers-11-02108-t002:** Nanosuspension evaluated for PPy immobilization on PDMS.

Nanosuspension Sample	Polyelectrolyte
PPy:PSS 2	PSS Mw 200,000
PPy:PSS 1	PSS Mw 70,000
PPy	None

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
