# Peer review of "Allylamine PECVD Modification of PDMS as Simple Method to Obtain Conductive Flexible Polypyrrole Thin Films"

_polymers, 2019, doi:10.3390/polym11122108_

Round 1

Reviewer 1 Report

This manuscript describes a simple method to obtain conductive flexible polypyrrole thin films, and it seems interesting. However, the lack of precision and logic is not matching the standard of a scientific journal like Polymers. Following are the detailed problems:

Figure 1 only shows the chemical structure of vinylamine, but the full text is talking about “allylamine grafting”.

In figure 2 A and lines 155 to 156, I severely doubt that the values of 3100 cm-1 and 1564 cm-1 are incorrect.

In figure 3 A, the image of transparent polymerization media should also be demonstrated.

In figure 6 A, C, and E, there are two kinds of rulers, 10 μm and 100 μm. So, which one is correct?

Moreover, there are so many grammatical errors! To give only a few examples: in lines 17 to 19, adding “the” before “films obtained” and removing the “and” before “as well as” are more appropriate. In line 25, are there any words lost before “is undeniable”?

Such mistakes should never show up even just submitting to a professional journal.

Reviewer 2 Report

You report a one-step method to obtain conductive polypyrrole thin films on
flexible substrates. To do that, substrates are modified through allylamine plasma grafting.

Overall all the information you gave and the extensive work you did, I found disappointing how the paper is missing crucial informations, please, read againg criptically and ask yourself about all the steps as you were an external user that would repeat the same experiment you did; you have still to work hard to present the paper.

Here following some point you must reconsider:

-In section 2, you don't cite any substrate, were have you bought the substrate? you say PDMS, does exist only one type, kind of PDMS?

-In section 2, how did you deposit the nanoparticle on top of the substrate? you cite in line 228 "previous section"...but were?

-In Figure 1 and 2 is missing the legend:

        Figure1 I guess that G is a molecule of gas, please use O2 and Ar, what is G? what are the red dots? and if you know how the grafting works (by creating a C-C bond via radical to a methyl group of the PDMS) try to write it, at least a little, in order that people know the molecular structure of you material.

        Figure2 This I think is a really nice figure but even if you know, the blue spagetti and the black balls, you have to explain what is what. on the last image, if there is also the PSS write the molecular structure...or at least PSS, to let people understand that is there

-revise please all the manuscript and try to make short phrases, from line 62 to line 66 there is only one comma, 27 to 29 no commas. Try to make short sentences, use the colon, semicolon etc.

-when you write you have to be coherent, use the comma or the point to separate decimals and hundreds but do not mix it: line 104 and 117 comma, 118 point; moreover in the same phrase line 117 you change using both.

-caption of figure 2, you don't say what is what, you have to think that people don't know your work, B and C are not explained, even if for you is obvious you should get the information only by reading the caption.

-Mw in line 115 and Mz in line 174, is the PSS changed?

-line 76, please give an introduction of Plasma, you make a nice introduction and suddently this technique pop out.

I'll appoint now some typos:

-line 59, Vaper?

-line 92, I guess here you should place Figure S1

-line 139 there is a spare letter "c".

-line 156, no vibration please, is a bending

-Figure 2A the FT-IR doesn't have any unit on Y

-line 210, is Figure S2 i guess

-line 212 plese check the bold on the bracket

-Figure 5A you use the slash or not to separe description and unit?

-line 295 two points at the end

-Figure 7D three stars up to the graph??

-line 311,312,313 not clear

Round 2

Reviewer 1 Report

The revised manuscript looks good and I would like to support it for publication.

Reviewer 2 Report

Dear authors,

this version results much better, this is the way you should have done since the beginning, moreover with the nice results you had the presentation is everytime crucial.

You have done a good work.

best regards